# Hand2 inhibits kidney specification while promoting vein formation within the posterior mesoderm

Elliot A Perens[1,2], Zayra V Garavito-Aguilar[1,3], Gina P Guio-Vega[3], Karen T Peña[3], Yocheved L Schindler[1], Deborah Yelon[1]*

[1]Division of Biological Sciences, University of California, San Diego, San Diego, United States; [2]Department of Pediatrics, School of Medicine, University of California, San Diego, San Diego, United States; [3]Departamento de Ciencias Biológicas, Facultad de Ciencias, Universidad de los Andes, Bogotá, Colombia

**Abstract** Proper organogenesis depends upon defining the precise dimensions of organ progenitor territories. Kidney progenitors originate within the intermediate mesoderm (IM), but the pathways that set the boundaries of the IM are poorly understood. Here, we show that the bHLH transcription factor Hand2 limits the size of the embryonic kidney by restricting IM dimensions. The IM is expanded in zebrafish *hand2* mutants and is diminished when *hand2* is overexpressed. Within the posterior mesoderm, *hand2* is expressed laterally adjacent to the IM. Venous progenitors arise between these two territories, and *hand2* promotes venous development while inhibiting IM formation at this interface. Furthermore, *hand2* and the co-expressed zinc-finger transcription factor *osr1* have functionally antagonistic influences on kidney development. Together, our data suggest that *hand2* functions in opposition to *osr1* to balance the formation of kidney and vein progenitors by regulating cell fate decisions at the lateral boundary of the IM.

*For correspondence: dyelon@ucsd.edu

**Competing interests:** The authors declare that no competing interests exist.

## Introduction

Organs arise from precisely defined territories containing progenitor cells with specific developmental potential. Distinct progenitor territories often abut one another, and communication at the interfaces between neighboring territories acts to refine their boundaries (*Dahmann et al., 2011*). This process delineates the final dimensions of each territory and, subsequently, influences the sizes of the derived organs. Boundary refinement is generally thought to be mediated by interplay between opposing inductive and suppressive factors (*Briscoe and Small, 2015*). In many cases, however, the identification of and interactions among these factors remain elusive.

Kidney progenitor cells originate from the intermediate mesoderm (IM), a pair of narrow bilateral stripes within the posterior mesoderm, flanked by lateral mesoderm that gives rise to vessels and blood and by paraxial mesoderm that gives rise to bone, cartilage, and skeletal muscle. The mechanisms that determine the dimensions of the stripes of IM are not fully understood. Several conserved transcription factors are expressed in the IM and are required for its development, including Lhx1/Lim1, Pax2, and Osr1/Odd1 (*Dressler and Douglass, 1992*; *James et al., 2006*; *Krauss et al., 1991*; *Toyama and Dawid, 1997*; *Tsang et al., 2000*; *Wang et al., 2005*). Studies in chick have indicated essential roles for the lateral mesoderm, paraxial mesoderm, and surface ectoderm in regulating the expression of these transcription factors (*James and Schultheiss, 2003*; *Mauch et al., 2000*; *Obara-Ishihara et al., 1999*). Furthermore, TGF-beta signaling acts in a dose-dependent manner to pattern the medial-lateral axis of the posterior mesoderm: for example, low levels of BMP signaling promote IM formation while high levels of BMP signaling promote lateral mesoderm formation

**eLife digest** The human body is made up of many different types of cells, yet they are all descended from one single fertilized egg cell. The process by which cells specialize into different types is complex and has many stages. At each step of the process, the selection of cell types that a cell can eventually become is increasingly restricted. The entire system is controlled by switching different genes on and off in different groups of cells. Balancing the activity of these genes ensures that enough cells of each type are made in order to build a complete and healthy body. Upsetting this balance can result in organs that are too large, too small or even missing altogether.

The cells that form the kidneys and bladder originate within a tissue called the intermediate mesoderm. Controlling the size of this tissue is an important part of building working kidneys. Perens et al. studied how genes control the size of the intermediate mesoderm of zebrafish embryos, which is very similar to the intermediate mesoderm of humans. The experiments revealed that a gene called *hand2*, which is switched on in cells next to the intermediate mesoderm, restricts the size of this tissue in order to determine the proper size of the kidney. Switching off the *hand2* gene resulted in zebrafish with abnormally large kidneys. Loss of *hand2* also led to the loss of a different type of cell that forms veins. These findings suggest that cells with an active *hand2* gene are unable to become intermediate mesoderm cells and instead go on to become part of the veins.

These experiments also demonstrated that a gene called *osr1* works in opposition to *hand2* to determine the right number of cells that are needed to build the kidneys. Further work will reveal how *hand2* prevents cells from joining the intermediate mesoderm and how its role is balanced by the activity of *osr1*. Understanding how the kidneys form could eventually help to diagnose or treat several genetic diseases and may make it possible to grow replacement kidneys from unspecialized cells.

(*Fleming et al., 2013*; *James and Schultheiss, 2005*). Beyond these insights, the pathways that set the boundaries of the IM and distinguish this territory from its neighbors are largely unknown.

Several lines of evidence have suggested that a carefully regulated refinement process is required to sharpen the boundary between the IM and the lateral mesoderm. In chick, mouse, and *Xenopus*, *Osr1* and *Lim1* are expressed in both the lateral mesoderm and the IM before becoming restricted to the IM, implying the existence of a mechanism that acts to exclude IM gene expression from the neighboring lateral territory (*Carroll and Vize, 1999*; *James et al., 2006*; *Mugford et al., 2008*; *Tsang et al., 2000*). Additional data have hinted at an antagonistic relationship between the IM and the blood and vessel lineages (*Gering et al., 2003*; *Gupta et al., 2006*): for example, overexpression of vascular and hematopoietic transcription factors (*tal1* and *lmo2*) induces ectopic vessel and blood specification while inhibiting IM formation (*Gering et al., 2003*). Along the same lines, zebrafish *osr1* morphants exhibit disrupted pronephron formation together with expanded venous structures (*Mudumana et al., 2008*). Despite these indications of interconnections between IM and vessel development, the network of factors that link these processes has not been fully elucidated.

Here, we establish previously unappreciated roles for the bHLH transcription factor Hand2 in both IM and vessel formation. Prior studies of Hand2 have focused on its functions in other tissues, including the heart, limb, and branchial arches (e.g. *Charité et al., 2000*; *Fernandez-Teran et al., 2000*; *Funato et al., 2009*; *Miller et al., 2003*; *Srivastava et al., 1997*; *Yanagisawa et al., 2003*; *Yelon et al., 2000*). Although Hand2 is also expressed in the posterior mesoderm (*Angelo et al., 2000*; *Fernandez-Teran et al., 2000*; *Srivastava et al., 1997*; *Thomas et al., 1998*; *Yelon et al., 2000*; *Yin et al., 2010*), its influence on patterning this tissue has not been extensively explored. Through both loss-of-function and gain-of-function studies, we find that *hand2* limits the size of the kidney by repressing IM formation while promoting venous progenitor formation. *hand2* is expressed laterally adjacent to the IM, and a set of venous progenitors arise at the interface between the *hand2*-expressing cells and the IM. Ectopic expression of IM markers within the *hand2*-expressing territory in *hand2* mutants suggests that *hand2* establishes the lateral boundary of the IM through direct inhibition of IM fate acquisition. Finally, genetic analysis indicates that *hand2* functions in opposition to *osr1* to control kidney dimensions. Together, our data demonstrate a novel

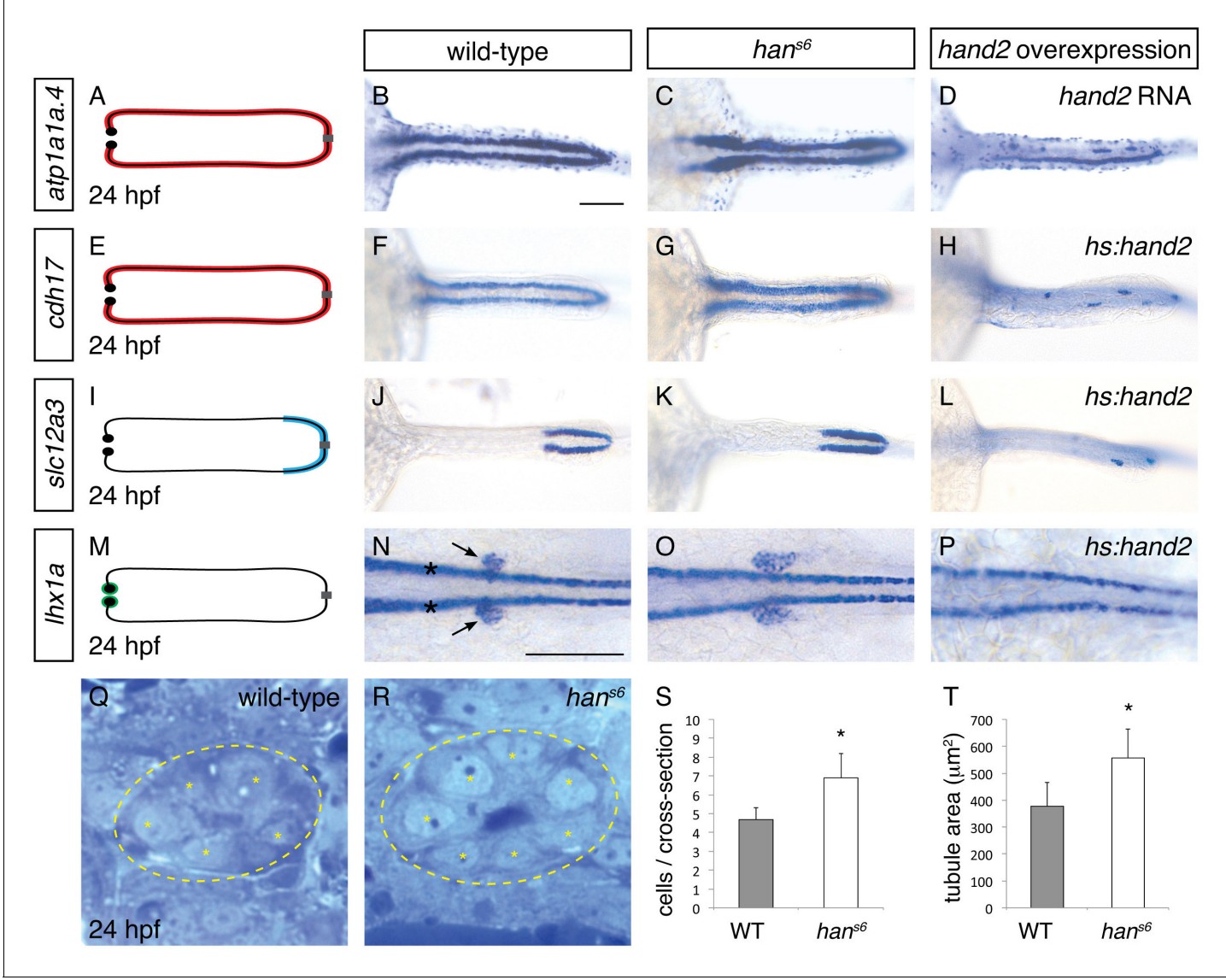

**Figure 1.** *hand2* inhibits pronephron formation. (A–P) Dorsal views, anterior to the left, of pronephron schematics (A, E, I, M), wild-type embryos (B, F, J, N), *han^s6* mutant embryos (C, G, K, O), and *hand2*-overexpressing embryos (D, injected with *hand2* mRNA; H, L, P, carrying *Tg(hsp70-hand2-2A-mcherry)*, abbreviated *hs:hand2*) at 24 hpf. In schematics (A, E, I, M), colored regions correspond to area of pronephron gene expression. In situ hybridization demonstrates that *atp1a1a*.4 (A–D) and *cdh17* (E–H) are expressed throughout the pronephron tubules, *slc12a3* (I–L) is expressed in the distal late segments of the pronephron tubules, and *lhx1a* (M–P) is expressed in the glomerular precursors (arrows, N), as well as overlying spinal neurons (asterisks, N). Compared to wild-type (B, F, J, N), gene expression is expanded in *han^s6* mutants (C, G, K, O) and reduced in *hand2*-overexpressing embryos (D, H, L, P). Of note, injection of a *hand2* translation-blocking morpholino caused effects on pronephron formation similar to those seen in *han^s6* mutants (data not shown). Scale bars represent 100 μm. (Q, R) Transverse sections through wild-type (Q) and *han^s6* mutant (R) pronephron tubules at 24 hpf. Dashed lines outline the tubule and asterisks indicate individual tubule cells. (S, T) Bar graphs indicate the average number of tubule cells per cross-section (S) and the average tubule area per cross-section (T) in wild-type and *han^s6* mutant embryos; error bars indicate standard deviation. Asterisks indicate statistically significant differences compared to wild-type ($p < 0.0001$, Student's t test; n = 18).

The following source data is available for figure 1:

**Source data 1.** Number of tubule cells per cross-section.
**Source data 2.** Area of pronephron tubule in cross-section.

mechanism for defining territory boundaries within the posterior mesoderm: *hand2* represses IM formation to establish its lateral boundary while promoting venous progenitor formation in this region. These important functions of Hand2 help to define the precise dimensions and components of the kidneys and vasculature. Moreover, these findings have implications for understanding the genetic basis of congenital anomalies of the kidney and urinary tract (CAKUT) and for developing new approaches in regenerative medicine.

## Results

### *hand2* limits pronephron dimensions by repressing pronephron formation

Our interest in the role of *hand2* during kidney development began with an observation arising from our previously reported microarray analysis of *hand2* mutants (*Garavito-Aguilar et al., 2010*). We compared gene expression profiles at 20 hr post fertilization (hpf) in wild-type embryos and *han[s6]* mutant embryos, which contain a deletion that removes the entire coding region of *hand2* (*Yelon et al., 2000*). Surprisingly, 11 of the 26 transcripts that were increased in *han[s6]* relative to wild-type were expressed in the pronephron, the embryonic kidney. (For a full list of differentially expressed genes, see *Garavito-Aguilar et al., 2010*.) We therefore sought to understand the effect of *hand2* function on the pronephron.

The six genes with the most elevated expression in *han[s6]* mutants (*Garavito-Aguilar et al., 2010*) are all expressed in the pronephron tubules. We examined the expression of two of these genes that are expressed throughout the tubules – *atp1a1a.4*, which encodes a subunit of the Na+/K+ ATPase (*Thisse et al., 2004*), and *cadherin17* (*cdh17*) (*Horsfield et al., 2002*) – and observed an increase in tubule width in *han[s6]* (*Figure 1A–C, E–G*). A similar increase in width of expression was seen for another gene upregulated in *han[s6]* but only expressed in a single tubule segment (*Wingert et al., 2007*), *slc12a3* (*Figure 1I–K*), which encodes the thiazide-sensitive sodium-chloride cotransporter solute carrier 12a3. Notably, in contrast to its widened expression, the anterior-posterior extent of *slc12a3* expression appeared unaltered. Last, by examining expression of *lhx1a* and *pax2a*, which mark glomerular precursors at 24 hpf (*O'Brien et al., 2011*), we found that the populations of glomerular precursors, like the tubules, were expanded in *han[s6]* (*Figure 1M–O* and Figure 8B,C). Thus, the elevated gene expression detected by microarray analysis in *han[s6]* mutants corresponded with broadened expression of genes throughout the pronephron.

To determine if this broadened expression was due to an increase in cell number, we next analyzed the structure of the pronephron in more detail. We found that the number of cells seen in cross-section of the tubule was increased in *han[s6]* (*Figure 1Q–S*). This increase in cell number was observed at multiple regions along the anterior-posterior axis, suggesting that the entire pronephron is expanded. Concomitant with this increase in the number of cells, the total tubule area in cross-section was increased in *han[s6]* mutants (*Figure 1T*). Together with the expanded expression of pronephron genes, these findings implicate *hand2* in inhibiting the accumulation of pronephron cells.

Does *hand2* simply prevent excessive expansion of the pronephron or is it potent enough to repress pronephron formation? To differentiate between these possibilities, we examined the effects of *hand2* overexpression on the pronephron. Injection of *hand2* mRNA inhibited pronephron formation, as assessed by *atp1a1a.4* expression (*Figure 1D*). To bypass any potential effects of overexpression on gastrulation, we also utilized our previously characterized *Tg(hsp70:FLAG-hand2-2A-mCherry)* transgenic line to drive *hand2* expression at the tailbud stage (*Schindler et al., 2014*). Heat shock-induced overexpression resulted in pronephron defects comparable to those caused by mRNA injection, as assessed by *atp1a1a.4*, *cdh17* and *slc12a3* expression (*Figure 1H,L* and *Figure 3A,B*). Furthermore, like tubule gene expression, *lhx1a* and *pax2a* expression in glomerular precursors was dramatically inhibited by *hand2* overexpression (*Figure 1P* and data not shown), again emphasizing the broad effect of *hand2* function on pronephron formation. In contrast to these effects of *hand2* overexpression on the pronephron, most other regions of expression of these markers, such as expression of *atp1a1a.4* in mucus-secreting cells and expression of *lhx1a* in spinal neurons, were unaffected (*Figure 1D,P*). Thus, the effect of *hand2* overexpression seems to reflect its particular impact on pronephron development, as opposed to a general influence of *hand2* on

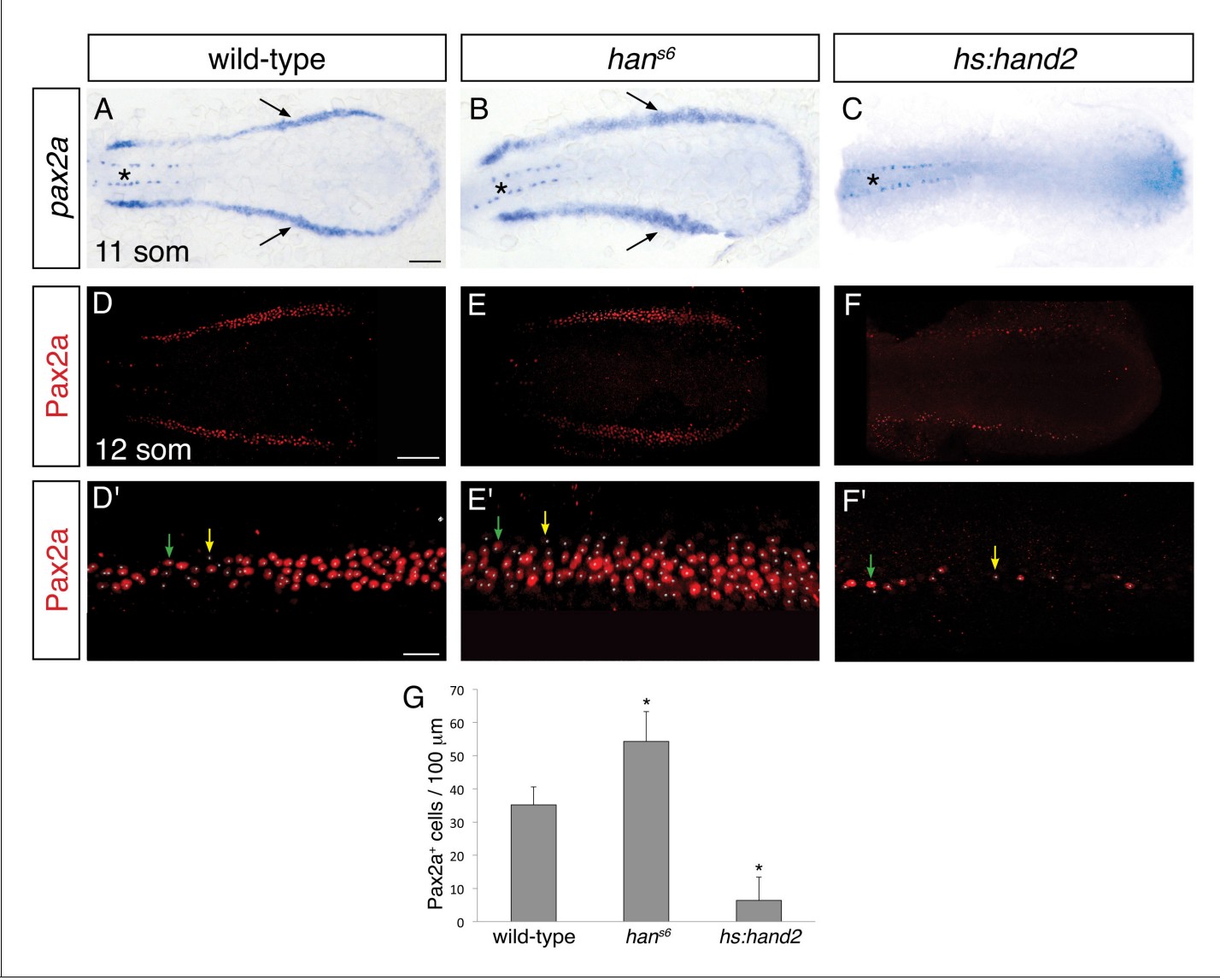

**Figure 2.** *hand2* inhibits IM production. (A–C) Dorsal views, anterior to the left, of the posterior mesoderm at the 11 somite stage. In situ hybridization depicts normal expression of *pax2a* in the IM (arrows) of wild-type embryos (A), widened expression in *han^s6* mutants (B), and lack of expression in *hand2*-overexpressing embryos (*hs:hand2*) (C). Expression in the spinal neurons (asterisk) is unaffected by altered *hand2* function. Scale bar represents 100 μm. (D–F) Pax2a immunofluorescence in the posterior mesoderm of wild-type (D), *han^s6* mutant (E), and *hs:hand2* (F) embryos at the 12 somite stage. Dorsal views, anterior to the left, are three-dimensional reconstructions of flat-mounted embryos from which the yolk and anterior tissues have been dissected away. Scale bar represents 100 μm. (D′–F′) Magnification of 250 μm long regions from (D–F) used for quantification of the number of Pax2a$^+$ cells in wild-type (D′), *han^s6* mutant (E′), and *hs:hand2* (F′) embryos. White dots indicate Pax2a$^+$ nuclei. Intensity of staining varied from strong (for example, green arrows) to weak (for example, yellow arrows). Scale bar represents 25 μm. (G) Bar graph indicates the average number of Pax2a$^+$ cells per 100 μm of IM in wild-type, *han^s6*, and *hs:hand2* mutant embryos; error bars indicate standard deviation. Asterisks indicate a statistically significant difference compared to wild-type ($p<0.0001$, Student's t test; n=13 for wild-type, n=10 for *han^s6*, and n=19 for *hs:hand2*).

The following source data and figure supplements are available for figure 2:

**Source data 1.** Pax2a$^+$ cells in wild-type, *han^s6*, and *hs:hand2* intermediate mesoderm.

**Figure supplement 1.** *hand2* inhibits IM production.

**Figure supplement 2.** Comparable proliferation in the IM of wild-type and *han^s6* mutant embryos.

**Figure supplement 2—source data 1.** Pax2a$^+$ pH3$^+$ cells in wild-type and *han^s6* intermediate mesoderm.

**Figure 3.** Pronephron development is susceptible to *hand2* overexpression prior to the 10 somite stage. (**A–E**) Dorsal views, anterior to the left, of in situ hybridization for *atp1a1a.4* at 24 hpf depict a range of severity of pronephron defects, ranging from absence of the pronephron (**A**) to unaffected (**E**). *Tg(hsp70:FLAG-hand2-2A-mCherry)* embryos were subjected to heat shock at the tailbud, 2 somite, 6 somite, or 10 somite stages, and the consequences on pronephron development were scored at 24 hpf. Percentages indicate the distribution of phenotypes produced by each treatment; the number of embryos examined is in the right-hand column. Heat shock at later stages resulted in more mild loss of *atp1a1a.4* expression in the tubule, and heat shock at the 10 somite stage did not disrupt *atp1a1a.4* expression in the tubule. Scale bar represents 100 μm.

the expression of each of these genes. Of note, however, we did observe a dramatic reduction in otic vesicle expression of both *atp1a1a.4* and *pax2a* (data not shown), suggesting a shared susceptibility to *hand2* overexpression in the otic vesicles and the pronephron. Taken together, our loss-of-function and gain-of-function studies show that *hand2* is necessary to constrain the size of the pronephron, likely through an ability to repress pronephron formation.

### *hand2* limits intermediate mesoderm dimensions by repressing intermediate mesoderm formation

We next sought to determine the origin of the effect of *hand2* on the pronephron, and we hypothesized that the pronephron defects observed in *han^{s6}* mutants might reflect a requirement for *hand2* to limit IM dimensions. Indeed, using two established IM markers, *pax2a* and *lhx1a*, we observed that the width of the IM was expanded in *han^{s6}* mutants and *hand2* morphants (*Figure 2A,B* and *Figure 2—figure supplement 1A,B*), at stages shortly after gastrulation (i.e. at the 6, 10 and 12 somite stages). In contrast, there was no substantial difference in the length of the IM seen in wild-type and *han^{s6}* mutant embryos (*Figure 2A,B*). Furthermore, compared to wild-type embryos, *han^{s6}* mutants typically had ~50% more Pax2a+ IM cells (*Figure 2D,E,G*). This accumulation of IM cells did not appear to be associated with heightened proliferation within the IM (*Figure 2—figure supplement 2*). Thus, our data suggest that *hand2* restricts the initial formation of the IM.

To determine whether *hand2* is sufficient to repress IM formation, we examined the effects of *hand2* overexpression. Both *hand2* mRNA injection and heat shock-induced overexpression of *hand2* resulted in loss of IM, as determined by *pax2a* and *lhx1a* expression (*Figure 2C* and *Figure 2—figure supplement 1C*). More specifically, *hand2*-overexpressing embryos exhibited phenotypes ranging from a significantly reduced number of Pax2a+ IM cells to a complete absence of Pax2a+ IM (*Figure 2F,G*). Taken together with our loss-of-function analyses, these results suggest that *hand2* limits the dimensions of the IM by repressing its initial specification.

To define the time window during which *hand2* overexpression can inhibit IM formation, we utilized *Tg(hsp70:FLAG-hand2-2A-mCherry)* embryos to induce *hand2* expression at different times after gastrulation (*Figure 3*). Unlike overexpression at the tailbud, 2 somite, or 6 somite stages (*Figure 3A–D*), overexpression at the 10 somite stage failed to inhibit pronephron development (*Figure 3E*). Furthermore, we found progressively less severe defects as heat shock induction was performed at successively later stages between tailbud and 10 somites. For example, while inhibition

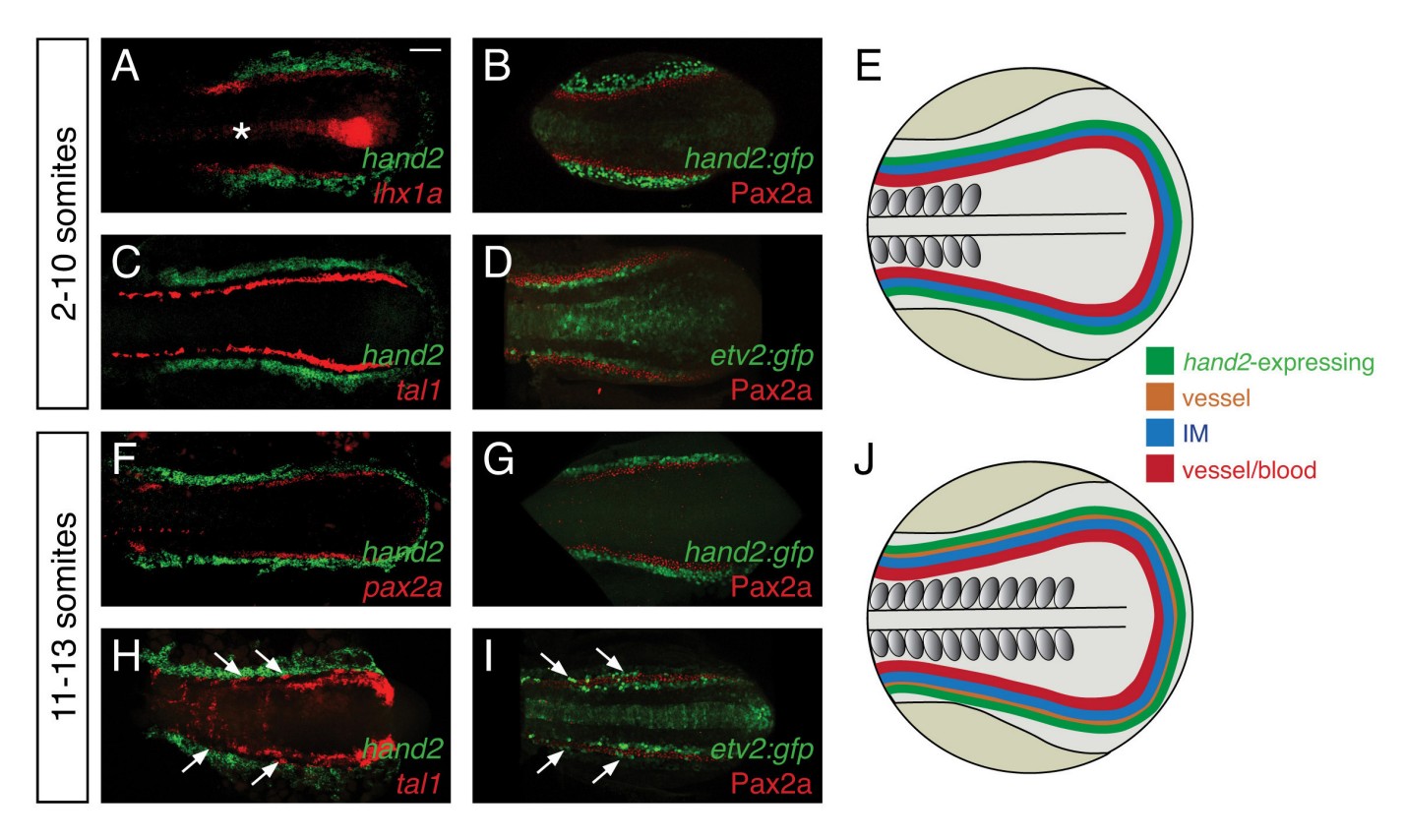

**Figure 4.** *hand2* expression in the posterior lateral mesoderm. (A–D, F–I) Two-color fluorescent in situ hybridization (A, C, F, H) and immunofluorescence (B, D, G, I) label components of the posterior mesoderm in dorsal views, anterior to the left, of three-dimensional reconstructions, as in *Figure 2D–E*. In embryos containing transgenes in which GFP expression is driven by the regulatory elements of *hand2* (B, G) or *etv2* (D, I), anti-GFP immunofluorescence was used to enhance visualization. *Tg(etv2:egfp)* expression was also observed in the midline neural tube, as previously reported (*Proulx et al., 2010*). Scale bar represents 100 μm. (E, J) Schematics depict posterior mesoderm territories, dorsal views, anterior to the left; *hand2*-expressing cells, IM, and medial vessel/blood progenitors are shown at 2–10 somites (E) and at 11–13 somites (J), together with lateral vessel progenitors. (A–D) *hand2* is expressed lateral to the IM at the 2–10 somite stages. Embryos shown are at the 10 somite stage; similar expression patterns were seen at earlier stages. (A, B) *hand2* is expressed lateral to the IM markers *lhx1a* (A) and Pax2a (B). (C) *hand2* is expressed lateral to *tal1*, a marker of blood and vessel progenitors; note the unlabeled gap between expression territories. (D) A marker of vessel progenitors, *Tg(etv2:egfp)*, lies medially adjacent to Pax2a. (F–I) Vessel progenitors arise at the interface between *hand2*-expressing cells and the IM at the 11–13 somite stages. (F, G) *hand2* is expressed lateral to the IM marker *pax2a*. (H) *hand2* is expressed lateral to a second territory of *tal1* expression; note the presence of lateral *tal1*-expressing cells (arrows) immediately adjacent to *hand2* expression. (I) The IM lies between two territories of *etv2* expression; the more lateral *etv2*-expressing cells (arrows) are lateral to the IM.

at the tailbud or 2 somite stages resulted in the loss of the majority of pronephric *atp1a1a.4* expression (*Figure 3A,B*), inhibition at the 6 somite stage resulted in only a mild reduction (*Figure 3D*). Overall, our analysis suggests that there is a *hand2*-sensitive phase of IM specification prior to the 10 somite stage.

### *hand2* is expressed beside the lateral boundary of the intermediate mesoderm

Considering the strong effect of *hand2* function on repressing IM formation, we sought to define the location of *hand2* expression relative to the IM. Prior studies had demonstrated bilateral *hand2* expression in the posterior mesoderm of zebrafish, mouse, chick, and *Xenopus* embryos (*Angelo et al., 2000*; *Fernandez-Teran et al., 2000*; *Srivastava et al., 1997*; *Thomas et al., 1998*; *Yelon et al., 2000*; *Yin et al., 2010*), but the precise localization of this expression relative to the IM had not been determined. During the *hand2*-sensitive phase of IM formation, we found *hand2* to be

expressed in bilateral regions immediately lateral to the IM (*Figure 4A,B*). At these stages, *hand2* was also expressed lateral to multiple markers of blood and vessel progenitors, including *etv2, tal1*, and *gata1* (*Figure 4C* and data not shown). However, a gap lies between these markers and the *hand2*-expressing cells, consistent with the IM residing between the lateral *hand2*-expressing cells and the medial blood and vessel progenitors (*Figure 4C,D*).

We also found that the relationship between *hand2* expression, the IM, and blood and vessel progenitors changed after the completion of the *hand2*-sensitive phase of IM formation. At approximately the 11 somite stage, a second, lateral population of vessel progenitors arises at the interface between the IM and the *hand2*-expressing cells (*Figure 4F–I*). These lateral vessel progenitors are likely to be venous progenitors, based on the results of a prior study that suggested that the medial, earlier-forming vessel progenitors contribute to the dorsal aorta and that the lateral, later-forming vessel progenitors contribute to the primary cardinal vein (*Kohli et al., 2013*).

This set of results defines the general location of *hand2* expression relative to other territories within the posterior mesoderm (*Figure 4E,J*). However, these observations do not exclude the possibility of transient overlapping expression at the boundaries of each territory (e.g. overlapping *etv2* and *pax2a* expression). Furthermore, we note that gene expression patterns are not necessarily uniform within each territory (e.g. *tal1* and *etv2* expression patterns are neither uniform nor equivalent within their territory [*Kohli et al., 2013*]). Nevertheless, our data suggest that, during the *hand2*-sensitive phase of IM formation, *hand2* may exert its repressive effect on IM formation by constraining the lateral boundary of the IM. Furthermore, our findings suggest the possibility of close interactions between *hand2*, the IM, and the lateral population of venous progenitors.

## *hand2* promotes lateral venous progenitor development in the posterior mesoderm

The appearance of lateral venous progenitor cells at the interface between the IM and the *hand2*-expressing cells raised the question of whether *hand2* regulates the development of these venous progenitors. To address this possibility, we first assessed the early expression of *etv2, tal1*, and *gata1* in the medial population of vessel and blood progenitors, and we observed no differences between wild-type and *han*[s6] mutant embryos at either the 6 or 10 somite stages (*Figure 5—figure supplement 1* and data not shown). In contrast, after the 11 somite stage, expression of both *etv2* and *tal1* in the lateral venous progenitor population was absent in *han*[s6] mutants (*Figure 5A,B,F,G*). Two-color fluorescent in situ hybridization confirmed the medial location of the remaining territory of *etv2* and *tal1* expression in *hand2* loss-of-function embryos (*Figure 5D,E,I,J*). Thus, the formation of the lateral venous progenitors, the appearance of which coincides with the end of the *hand2*-responsive phase of IM formation, requires *hand2*.

To gain insight into whether *hand2* directly promotes the formation of vessel progenitors, we assessed the consequences of *hand2* overexpression. Overexpression of *hand2* resulted in an expansion of both *etv2* and *tal1* expression in the posterior mesoderm (*Figure 5C,H*). In contrast, *hand2* overexpression resulted in a reduction of *gata1* expression (*Figure 5M*). Thus, in the posterior mesoderm, *hand2* overexpression can inhibit the development of some tissues, such as IM and blood, while promoting vessel progenitor formation.

To understand the consequences of the influence of *hand2* function on the lateral venous progenitors, we examined the vasculature in *han*[s6] mutants. As noted above, a prior lineage tracing study suggested that the lateral venous progenitors contribute to the posterior cardinal vein (*Kohli et al., 2013*). Furthermore, this study found that inhibition of *etv2* function at the time when these lateral cells arise results in loss of expression of *mannose receptor c1a (mrc1a)*, a marker of the cardinal vein (*Kohli et al., 2013*). Similarly, *han*[s6] mutants lack *mrc1a* and *flt4* expression in the posterior cardinal vein, suggesting a defect in venous development (*Figure 6A,B,D,E*). Using *Tg(flk1:ras-mcherry)* to label the entire vasculature, we identified the presence of the posterior cardinal vein in *han*[s6] mutants (*Figure 6J–L*), but found that their caudal venous plexus failed to be properly remodeled (*Figure 6M–O*). In contrast to these changes in the venous system, *flt4* and *ephrin-b2a (efnb2a)* expression in the dorsal aorta, as well as *mrc1a* expression in the posterior blood island, appeared grossly unaffected by *hand2* loss-of-function (*Figure 6B,E,H*). Meanwhile, consistent with the ability of *hand2* to promote vessel progenitor formation, *hand2* overexpression increased vascular expression of *mrc1a, flt4*, and *efnb2a* (*Figure 6C,F,I*).

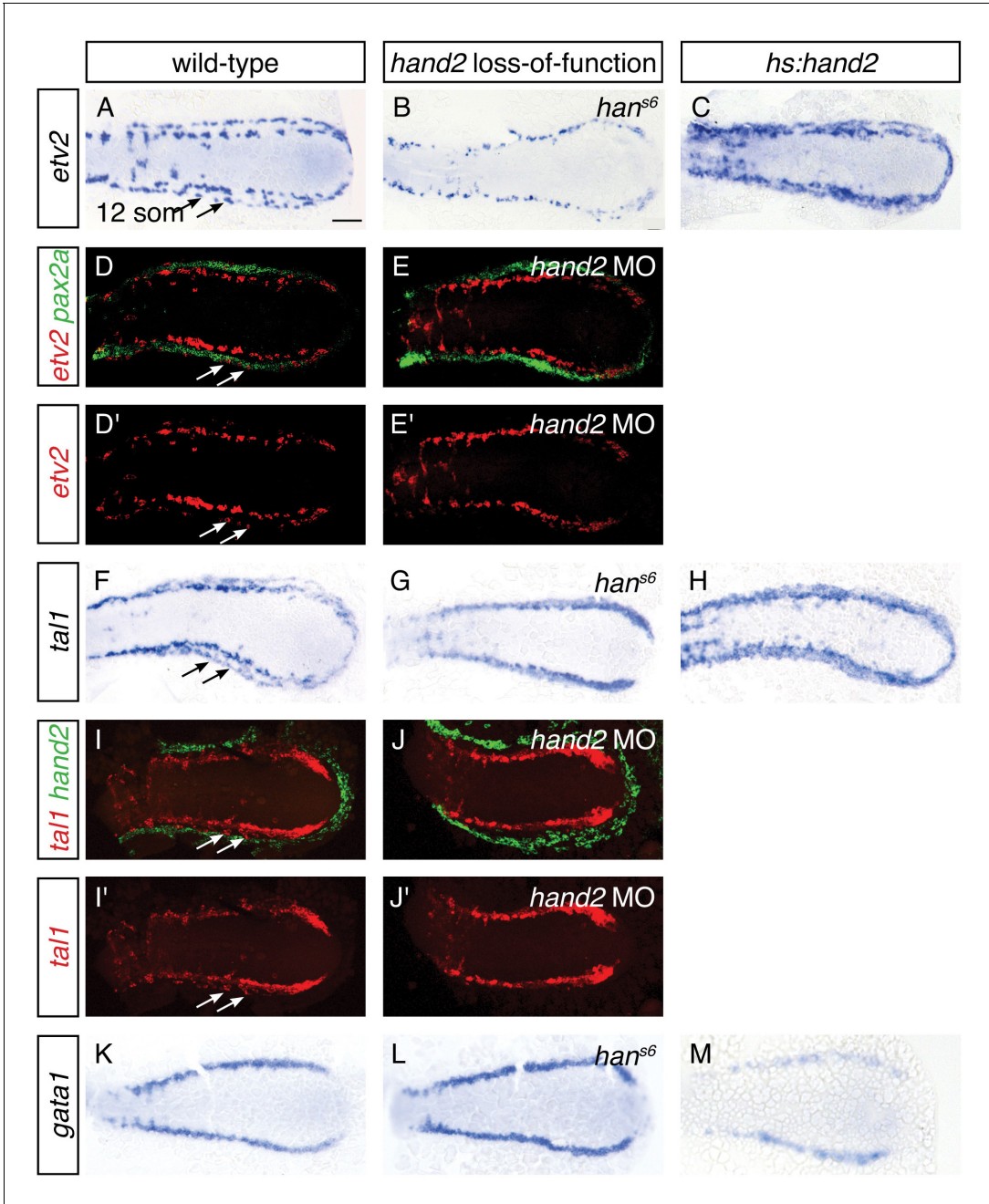

**Figure 5.** *hand2* promotes vessel progenitor development. (A–C, F–H, K–M) In situ hybridization depicts *etv2* (A–C), *tal1* (F–H) and *gata1* (K–M) expression in wild-type (A, F, K), *han^s6* mutant (B, G, L) and *hand2*-overexpressing (C, H, M) embryos; dorsal views, anterior to the left, at the 12 somite stage. (A, F) *etv2* and *tal1* are expressed in relatively medial and lateral (arrows) territories on each side of the wild-type embryo. In *han^s6* embryos (B, G), only the medial territory is present. In *hand2*-overexpressing embryos (C, H), expression of both *etv2* and *tal1* is increased, but it is not possible to distinguish whether this represents an increase in the medial or the lateral vessel progenitor populations, since no markers exist that distinguish these two groups of progenitors. (K–M) *gata1* expression is equivalent in wild-type (K) and *han^s6* (L) embryos, but it is decreased in *hand2*-overexpressing embryos (M). Scale bar represents 100 µm. (D, E, I, J) Fluorescent in situ hybridization depicts the relationship of *etv2* and *pax2a* expression in wild-type (D, D') and *hand2* morphant (*hand2* MO) embryos (E, E'), and the relationship of *tal1* and *hand2* expression in wild-type (I, I') and *hand2* MO (J, J') embryos; dorsal views, anterior to the left, at the 12 somite stage. Medial and lateral (arrows) territories of *etv2* expression flank *pax2a* in wild-type embryos (D, D'). The lateral territory of expression is absent in *hand2* morphants (E, E'). The lateral territory (arrows) of *tal1* expression is located at the medial border of *hand2* expression in wild-type embryos (I, I'), but is absent in *hand2* morphants (J, J'). Note that we observed variable thickness of *tal1* expression in its medial territory of expression.

*Figure 5 continued on next page*

*Figure 5 continued*

The following figure supplement is available for figure 5:

**Figure supplement 1.** Presence of medial vessel and blood progenitors is unaffected in *han*[s6] mutants.

Overall, these studies demonstrate a previously unappreciated role for *hand2* in regulating trunk vasculature development. More specifically, our data indicate that *hand2* is required for the successful execution of certain aspects of venous differentiation, including the expression of characteristic venous markers and the remodeling of the caudal venous plexus. Combined with prior lineage tracing studies and *etv2* loss-of-function analysis (**Kohli et al., 2013**), our findings suggest that *hand2* implements these functions by promoting the development of the lateral venous progenitors. We cannot, however, rule out the possibility that the roles of *hand2* in venous progenitor formation and venous differentiation are independent of one another, such that the differentiation defects represent a later influence of *hand2* on vascular development.

## *hand2* loss-of-function results in increased expression of intermediate mesoderm markers in cells that normally express *hand2*

The lateral localization of *hand2* expression and the absence of lateral venous progenitor markers when *hand2* function is lost highlighted the possibility that *hand2* impacts the IM at its lateral border. We therefore wanted to determine how the expanded IM in *han*[s6] mutants relates to the interface between the IM and the *hand2*-expressing cells. Do the extra IM cells seem to emerge at this interface? Do they appear to arise from cells that express *hand2*, or, alternatively, at the expense of *hand2*-expressing cells?

To differentiate among these possibilities, we used two methods to identify the *hand2*-expressing territory in *hand2* loss-of-function embryos. We labeled *hand2*-expressing cells through in situ hybridization in embryos injected with a *hand2* morpholino (**Figure 7A,B**), and we took advantage of the non-rescuing BAC transgene *Tg(hand2:EGFP)* (**Kikuchi et al., 2011**) to interrogate *han*[s6] mutants (**Figure 7C,D**). Of note, each of these approaches indicated heightened intensity of *hand2* expression in *hand2* loss-of-function embryos (**Figure 7A,B** and **Figure 7—figure supplement 1**), suggesting the possibility that *hand2* acts to inhibit its own expression. Importantly, however, we observed comparable dimensions of the *hand2* expression territory in wild-type and morphant embryos (**Figure 7A,B**). More specifically, when using *Tg(hand2:EGFP)*, we found no significant difference between the numbers of GFP⁺ cells in the posterior mesoderm of wild-type and *han*[s6] mutant embryos (**Figure 7C–E**). Thus, it seems that the expansion of the IM in *han*[s6] mutants does not come at the expense of the formation of cells that normally express *hand2*.

We next assessed whether the increased number of IM cells in *han*[s6] mutants could arise at least in part from aberrant IM marker expression in cells that normally express *hand2*. Careful examination of wild-type embryos carrying the *Tg(hand2:EGFP)* transgene revealed that ~6% of the GFP⁺ cells in their posterior mesoderm also displayed the IM marker Pax2a (**Figure 7C,E**). In dramatic contrast, ~60% of the GFP⁺ cells in the *han*[s6] mutant posterior mesoderm were also Pax2a⁺ (**Figure 7D,E**). Intriguingly, the GFP⁺ Pax2a⁺ cells were typically observed at or near the border where the *hand2*-expressing cells meet the IM (**Figure 7C,D**). These data suggest that *hand2* limits IM dimensions by constraining the lateral boundary of the IM through repression of IM gene expression within *hand2*-expressing cells.

## Functionally antagonistic roles for *hand2* and *osr1* in pronephron development

The expansion of Pax2a expression in the *hand2*-expressing territory in *han*[s6] mutants suggests that factors that promote IM development are present within *hand2*-expressing cells. We hypothesized that *osr1* could be one of these factors, because of the correspondence between the *hand2* and *osr1* expression patterns and loss-of-function phenotypes. First, *osr1* was previously reported to be expressed in the lateral posterior mesoderm, adjacent to the IM (**Mudumana et al., 2008**). Through direct comparison of *hand2* and *osr1* expression patterns, we found that the genes are co-expressed in the lateral posterior mesoderm (**Figure 8A**). Additionally, prior studies demonstrated that loss of

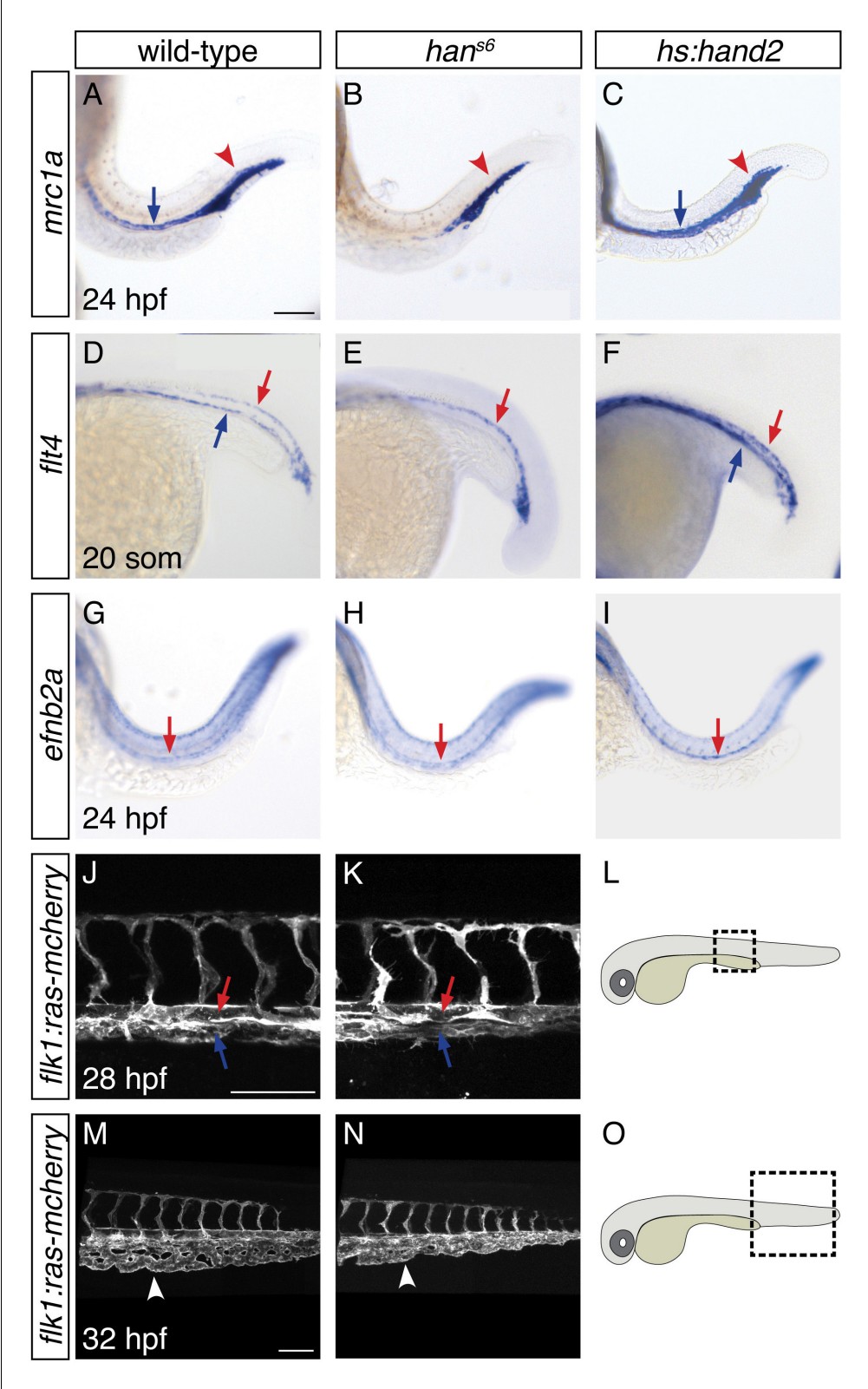

**Figure 6.** *hand2* promotes proper vein formation. (**A–I**) In situ hybridization depicts expression of *mrc1a* (**A–C**), *flt4* (**D–F**) and *efnb2a* (**G–I**) in wild-type (**A, D, G**), *han*s6 mutant (**B, E, H**) and *hand2*-overexpressing (**C, F, I**) embryos; lateral views, anterior to the left, at 24 hpf (**A–C, G–I**) and the 20 somite stage (**D–F**). (**A–C**) *mrc1a* expression in the posterior cardinal vein (blue arrow) was present in wild-type (**A**), absent in *han*s6 mutant (**B**), and increased in *hand2*-overexpressing (**C**) embryos. Expression in the posterior blood island (red arrowhead) was grossly unaffected. (**D–F**) *flt4* expression in the

*Figure 6 continued on next page*

*Figure 6 continued*

posterior cardinal vein (blue arrow) was present in wild-type (D), absent in *han^s6* mutant (E), and increased in *hand2*-overexpressing (F) embryos. Expression in the dorsal aorta (red arrow) was grossly unaffected. (G–I) *efnb2a* expression in the dorsal aorta (red arrow) was present in wild-type (G), grossly unaffected in *han^s6* mutant (H), and slightly increased in *hand2*-overexpressing (I) embryos. (J–O) Lateral views of three-dimensional reconstructions of *Tg(flk1:ras-mcherry)* expression in the vasculature of wild-type (J, M) and *han^s6* mutant (K, N) embryos at 28 hpf (J, K) and 32 hpf (M, N). Pictured region in (J, K) is the area of the trunk boxed in the schematic (L); pictured region in (M, N) is the area of the tail boxed in the schematic (O). Both posterior cardinal vein (blue arrow) and dorsal aorta (red arrow) were present in wild-type (J) and *han^s6* mutant (K) embryos. In contrast to wild-type (M), the caudal venous plexus (arrowhead) fails to undergo proper remodeling in *han^s6* mutants (N). Scale bars represent 100 μm.

*osr1* function results in phenotypes opposite to those caused by loss of *hand2*: in contrast to the expanded pronephron and defective venous vasculature in *han^s6* mutants, *osr1* morphants were found to exhibit inhibited pronephron development together with expansion of venous structures (*Mudumana et al., 2008*; *Tena et al., 2007*).

To assess the functional relationship between *hand2* and *osr1*, we investigated double loss-of-function embryos. Dramatically, removal of *hand2* function in the setting of *osr1* loss-of-function largely restored normal pronephron development. This interaction was most notable in the glomerular precursor population (*Figure 8B–E*). While this population was expanded in *hand2* mutants, it was absent in embryos injected with an *osr1* morpholino (*Figure 8B–D*). In *hand2* mutants injected with the *osr1* morpholino, however, the precursor population appeared comparable to wild-type (*Figure 8E*). Consistent with prior studies (*Mudumana et al., 2008*), *osr1* morpholino injection had a less dramatic effect on the pronephron tubule than on the glomerulus (*Figure 8D,H*). Nevertheless, a comparable interaction between *hand2* and *osr1* was found in tubule formation (*Figure 8F–I*). While *han^s6* mutants had wide tubules and many *osr1* morphants had disrupted tubules, *han^s6* mutantsinjected with *osr1* morpholino displayed relatively normal tubules.

We wondered whether the observed genetic interaction between *hand2* and *osr1* might reflect roles for these genes in regulating one another's expression. We found, however, that loss of *hand2* function did not seem to affect the intensity or dimensions of *osr1* expression (*Figure 8—figure supplement 1A,B*). Similarly, *osr1* loss-of-function did not appear to alter *hand2* expression (*Figure 8—figure supplement 1C,D*). Therefore, *hand2* and *osr1* appear to affect downstream gene expression independently, without affecting one another's expression. Furthermore, our data suggest that *hand2* and *osr1* function in parallel genetic pathways that act in opposition in order to regulate the precise dimensions of the pronephron. More generally, our findings implicate *hand2* within a genetic network that regulates progenitor cell fates at the lateral edge of the IM, at the interface with the *hand2*- and *osr1*-expressing territory.

## Discussion

Together, our data present a new perspective on how the posterior mesoderm is divided into defined territories that create distinct tissues. Importantly, our work provides the first evidence that *hand2* restricts the size of the kidney and promotes vein development. Specifically, our data indicate that *hand2* limits the specification of pronephric progenitors by restraining the lateral boundary of the IM. Furthermore, at the end of the *hand2*-sensitive phase of IM formation, *hand2* supports the formation of venous progenitors just beyond the lateral edge of the IM, in the same region where it represses IM development. Finally, our data reveal a novel genetic interaction in which the inhibitory function of *hand2* is balanced by the inductive role of *osr1* during the establishment of IM dimensions.

Our findings highlight the importance of regulating the refinement of the lateral boundary of the IM. It is intriguing to consider how *hand2*, given its expression in the lateral posterior mesoderm, could influence the restriction of the adjacent IM. Perhaps *hand2* acts cell-autonomously to inhibit IM specification; the ectopic appearance of Pax2a^+ cells within the *hand2*-expressing territory in *hand2* mutants makes this an attractive model. Alternatively, *hand2* could have a non-autonomous influence on IM dimensions, perhaps by regulating the production of a lateralizing signal, the absence of which would result in the lateral expansion of IM gene expression. In this scenario, it is tempting to speculate that the role of *hand2* could interface with the BMP signaling pathway,

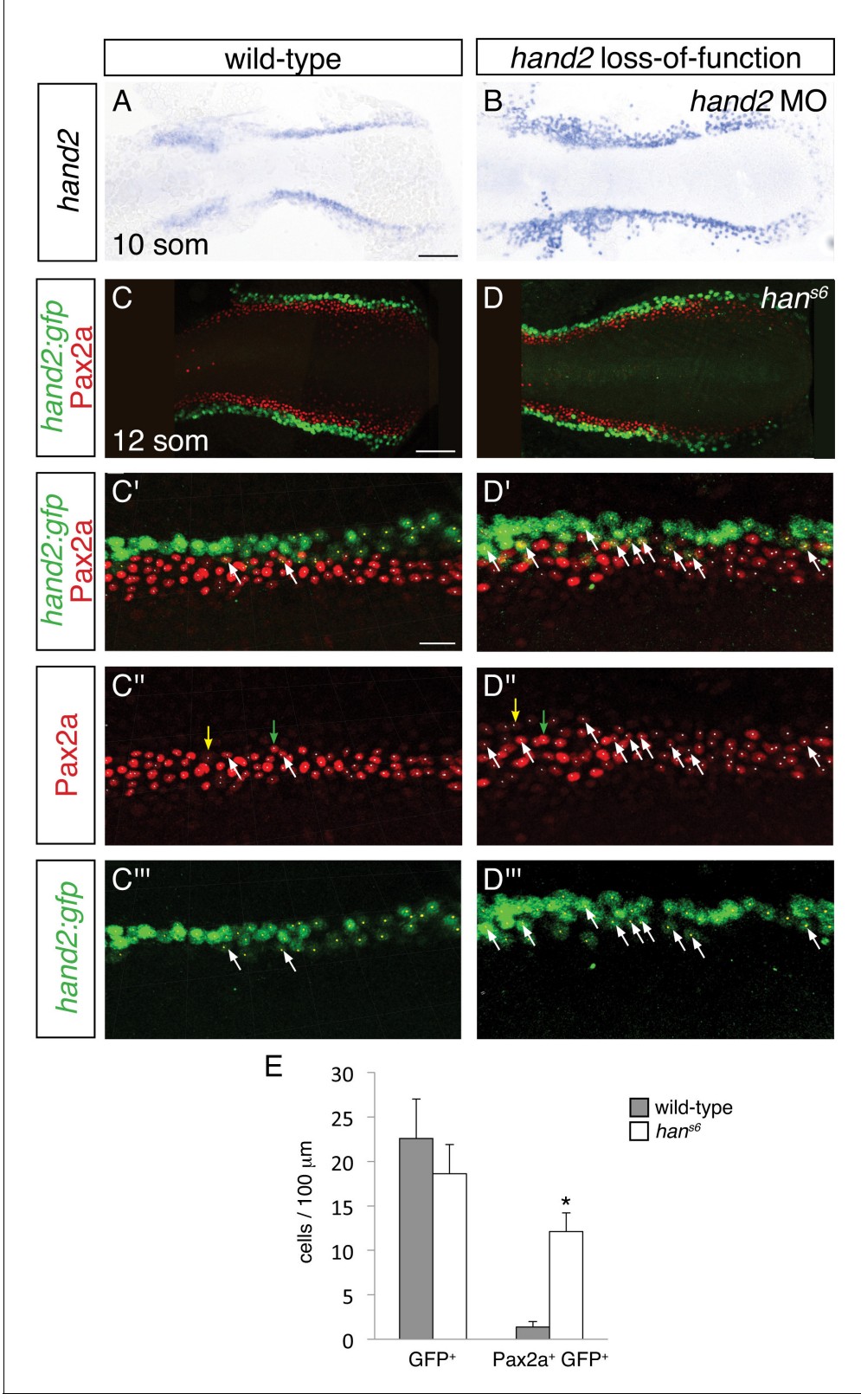

**Figure 7.** Increased presence of Pax2a in *hand2*-expressing cells of *hand2* mutants. (A–B) In situ hybridization depicts presence of *hand2* expression in *hand2* morphants (B); dorsal views, anterior to the left, at the 10 somite stage. There is no evident loss of *hand2*-expressing cells in *hand2* morphants; moreover, *hand2* expression levels appear higher in the context of *hand2* loss-of-function. Scale bar represents 100 μm. (C–D) Immunofluorescence for Pax2a and GFP in wild-type (C) and *han^s6* mutant (D) embryos, both carrying *Tg(hand2:EGFP)*; dorsal views, anterior to the left, of three-dimensional

*Figure 7 continued on next page*

Figure 7 continued

reconstructions at the 12 somite stage, as in *Figure 4G*. (C'–D'') Magnification of 250 μm long regions from (C) and (D) used for quantification of the numbers of GFP+ and Pax2a+ cells in wild-type and *han^s6* mutant embryos. Yellow dots indicate GFP+ cells, white dots indicate Pax2a+ nuclei, and examples of GFP+ Pax2a+ cells are indicated by white arrows. Intensity of Pax2a+ staining varied from strong (for example, green arrows) to weak (for example, yellow arrows). Scale bars represent 100 μm (C, D) and 25 μm (C'–D'''). (E) Bar graph indicates the average numbers of GFP+ cells and GFP+ Pax2a+ cells per 100 μm of IM in wild-type and *han^s6* mutant embryos; error bars indicate standard deviation. Asterisk indicates a statistically significant difference compared to wild-type (p<0.0001, Student's t test; n = 13 for wild-type and n = 10 for *han^s6*).

The following source data and figure supplement are available for figure 7:

**Source data 1.** GFP+ and Pax2a+ GFP+ cells in wild-type and *han^s6* intermediate mesoderm.

**Figure supplement 1.** Increased *hand2:gfp* expression in *han^s6*.

considering its prior implication in posterior mesoderm patterning (*James and Schultheiss, 2005*). Wherever its precise location of action may be, this function of *hand2* – preventing lateral *hand2+-osr1+* cells from acquiring more medial characteristics, such as expression of *pax2a* or *lhx1a* –suggests a possible mechanism for distinguishing IM gene expression from that of neighboring lateral

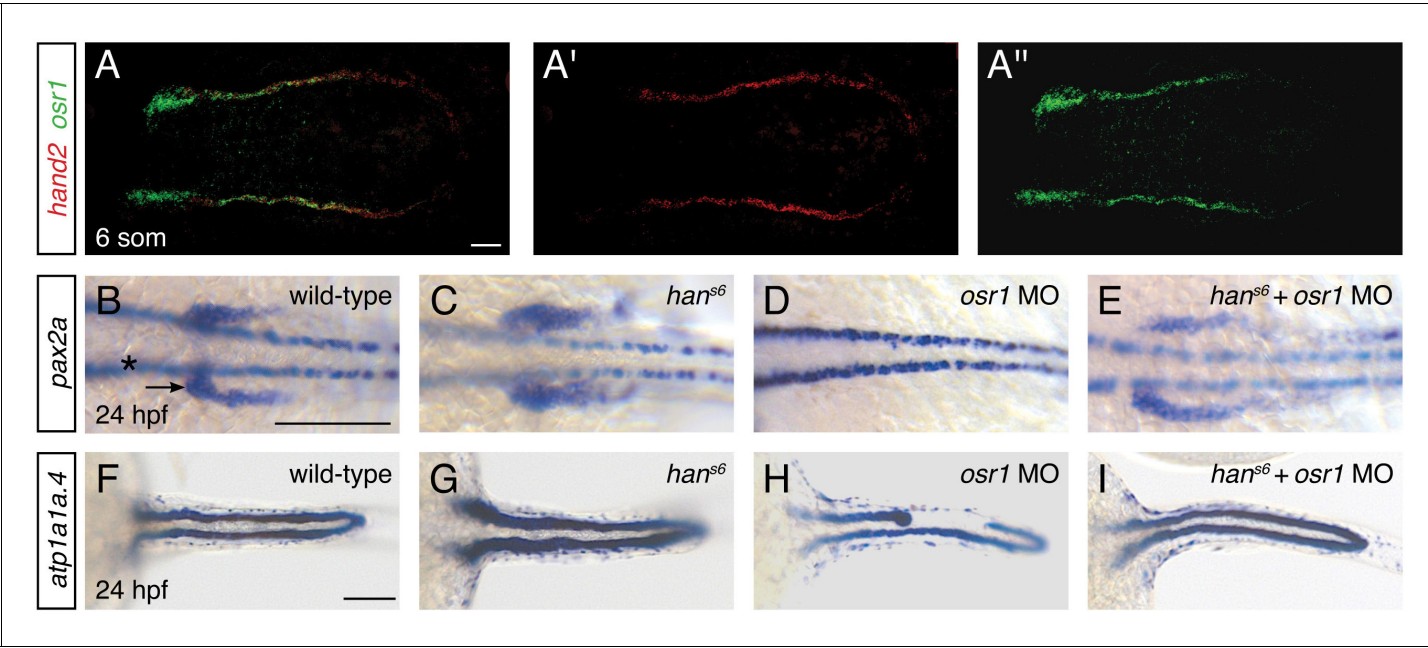

**Figure 8.** *hand2* and *osr1* act in opposing, parallel pathways to regulate pronephron development. (A–A'') Fluorescent in situ hybridization depicts overlap (A) of *hand2* (A') and *osr1* (A'') expression in wild-type embryos; dorsal views, anterior to the left, of three-dimensional reconstructions at the 6 somite stage. (B–I) In situ hybridization depicts *pax2a* (B–E) and *atp1a1a.4* (F–I) expression at 24 hpf in wild-type embryos (B, F), *han^s6* mutant embryos (C, G), *osr1* morphant (*osr1* MO) embryos (D, H) and *han^s6* mutant embryos injected with *osr1* morpholino (*han^s6* + *osr1* MO) (E, I); dorsal views, anterior to the left. (B–E) Compared to wild-type (B), *pax2a* expression in the glomerular and neck precursors (arrow) was expanded in 100% of *han^s6* mutants (C, n=11), absent (48%) or reduced (48%) in *osr1* morphants (D, n=25), and relatively normal in *han^s6* + *osr1* MO embryos (E, n=11). While the extent of marker expression was generally comparable to wild-type in the double loss-of-function embryos, the stereotypic patterning of this population was often somewhat disrupted. Expression in overlying spinal neurons (asterisk) was unaffected. (F–I) Compared to wild-type (F), *atp1a1a.4* expression in the pronephric tubules was wide in 85% of *han^s6* mutants (G, n=13), while many *osr1* morphants (H, n=133) had tubules with shortened anterior expression (18%) or tubules with segmental losses (35%), and 47% of *osr1* morphants had a wild-type appearance. 85% of *han^s6* + *osr1* MO embryos (I, n=46) resembled wild-type, whereas 11% had a shortened anterior tubule and 4% had segmental losses in the tubules. Scale bars represent 100 μm.

The following figure supplement is available for figure 8:

**Figure supplement 1.** Expression patterns of *osr1* and *hand2* appear unaffected by loss of each other's function.

territories, a distinct process required during both in vivo (*James et al., 2006*) and in vitro (*Takasato et al., 2014*) IM development.

Additionally, our data imply a close connection between the specification of the IM and the formation of the lateral venous progenitor population. Are the roles of *hand2* during IM and venous development coupled to each other, or do these represent two independent functions of *hand2*? We can envision a model in which the lateral venous progenitor cells emerge from a *hand2*-expressing lineage, and *hand2* influences a cell fate decision between IM and venous identities. In particular, we suggest that *hand2* normally inhibits IM specification within a portion of the *hand2*-expressing territory, while also promoting the differentiation of these same *hand2*-expressing cells into lateral venous progenitors. On the other hand, it remains feasible that *hand2* could have completely separable effects on the kidney and vein lineages. Currently, the lineage relationships between the IM, the lateral venous progenitors, and the *hand2*-expressing lateral mesoderm remain undetermined; future work on the generation of appropriate tissue-specific lineage tracing tools will facilitate progress toward the resolution of these open questions.

In future studies, it will also be valuable to elucidate the effector genes that act downstream of Hand2 in the posterior mesoderm. It is plausible that Hand2 could act directly to repress transcription of IM genes, especially considering that ChIP-seq analysis in the mouse limb bud has revealed Hand2-binding peaks associated with *Pax2* and *Lhx1a* (*Osterwalder et al., 2014*). Alternatively, Hand2 could repress IM genes without necessarily engaging Hand2-binding sites in their regulatory regions, since Hand2 can influence transcription through DNA-binding-independent mechanisms (*Funato et al., 2009*; *Liu et al., 2009*). Of course, Hand2 could also regulate IM gene expression indirectly. It will be particularly interesting to determine whether the Hand2 targets that influence the IM are also relevant to venous progenitor formation. Along those lines, it is noteworthy that our findings have demonstrated a difference between the functions of *hand2* during vessel progenitor development in the anterior and posterior mesoderm: while we have previously shown that *hand2* overexpression inhibits the expression of vessel progenitor markers in the anterior mesoderm (*Schindler et al., 2014*), here we observed expanded expression of the same markers in the posterior mesoderm of *hand2*-overexpressing embryos. These distinct responses to *hand2* activity in the anterior and posterior mesoderm may reflect regional differences in in the utilization of different bHLH binding partners or in the implementation of the different molecular functions ascribed to Hand2 (*Firulli et al., 2005*; *Funato et al., 2009*; *Liu et al., 2009*; *McFadden et al., 2002*; *Schindler et al., 2014*). Future studies will need to distinguish among these possibilities in order to define how Hand2 functions in different embryonic contexts.

By demonstrating that *hand2* functions in opposition to *osr1*, our data provide the first key step toward incorporating *hand2* into the genetic regulatory network that patterns the posterior mesoderm. It is important to note that prior studies have suggested a non-autonomous role for *osr1* during IM development (*Mudumana et al., 2008*). The endoderm is expanded in *osr1* morphants, and mutations that disrupt endoderm development abrogate the pronephron defects caused by *osr1* loss-of-function (*Mudumana et al., 2008*; *Terashima et al., 2014*). In contrast, we suspect that *hand2* affects IM formation from its position in the lateral mesoderm, rather than from the endoderm. We observe ectopic Pax2a[+] cells emerging within *hand2*-expressing mesoderm in *hand2* mutants; additionally, whereas *osr1* morphants have an expansion of the endoderm (*Mudumana et al., 2008*; *Terashima et al., 2014*), *han*[s6] mutants exhibit normal amounts of endoderm (*Wendl et al., 2007*; *Yelon et al., 2000*). Regardless of whether *hand2* and *osr1* both function within the lateral posterior mesoderm, our data suggest that these two genes control antagonistic and parallel genetic pathways that balance each other's influence in order to regulate the size of the IM.

It will be interesting to determine whether the relationship between *hand2* and *osr1* function is broadly conserved. Loss-of-function and gain-of-function studies in mouse, chick, and *Xenopus* have demonstrated a role for *Osr1* in IM and kidney development through focused analyses of histology and gene expression patterns (*James et al., 2006*; *Mugford et al., 2008*; *Tena et al., 2007*; *Wang et al., 2005*). In contrast, while mice lacking *Hand2* have been examined for developmental defects in some posterior structures, such as the limb buds and enteric nervous system (e.g. *Charité et al., 2000*; *Lei and Howard, 2011*), the influence of *Hand2* on IM and kidney development has not yet been carefully interrogated. Moreover, future investigations in mouse may need to evaluate potential redundancy between the functions of *Hand2* and *Hand1*: both of these genes are

expressed in the murine lateral posterior mesoderm (*Thomas et al., 1998*), whereas zebrafish have only a single Hand gene.

Overall, our new insights into the genetic regulation of IM specification have the potential to impact our understanding of the origins and treatment of kidney disease. Prior studies have suggested a genetic component to the origins of congenital anomalies of the kidney and urinary tract (CAKUT) (*Bulum et al., 2013*; *Vivante et al., 2014*). In some cases of CAKUT, disease-causing mutations have been identified, many of which disrupt genes that influence early kidney development, including mutations in *PAX2* and *OSR1* (*Madariaga et al., 2013*; *Thomas et al., 2011*; *Zhang et al., 2011*). By implicating *hand2* in the control of IM specification, we enrich the gene regulatory network that is relevant to the etiology of CAKUT. In this regard, it is interesting to note that there are case reports of chromosomal duplications containing *HAND2* in patients with renal hypoplasia (*Otsuka et al., 2005*). Furthermore, the identification of genes that control IM specification can inspire new directions in the design of stem cell technologies and regenerative medicine strategies. Alteration of *HAND2* function may allow for enhancement of current protocols for generating kidney progenitor cells in vitro (*Kumar et al., 2015*; *Mae et al., 2013*; *Taguchi et al., 2014*; *Takasato et al., 2014*; *Takasato et al., 2015*), potentially facilitating the requisite control that will be essential for future deployment in patient applications.

# Materials and methods

## Zebrafish

We generated embryos by breeding wild-type zebrafish, zebrafish heterozygous for the *hand2* mutant allele *han^{s6}* (*Yelon et al., 2000*) (RRID: ZFIN_ZDB-GENO-071003-2), and zebrafish carrying *Tg(hand2:EGFP)^{pd24}* (*Kikuchi et al., 2011*) (RRID: ZFIN_ZDB-GENO-110128-35), *Tg(hsp70:FLAG-hand2-2A-mCherry)^{sd28}* (*Schindler et al., 2014*) (RRID: ZFIN_ZDB-GENO-141031-2), *Tg(flk1:ras-mcherry)^{s896}* (*Chi et al., 2008*) (RRID: ZFIN_ZDB-GENO-081212-2), or *Tg(etv2:egfp)^{ci1}* (*Proulx et al., 2010*) (RRID: ZFIN_ZDB-GENO-110131-58). For induction of heat shock-regulated expression, embryos were placed at 37°C for 1 hr and then returned to 28°C. Following heat shock, transgenic embryos were identified based on mCherry fluorescence; nontransgenic embryos were analyzed as controls. All heat shocks were performed at tailbud stage unless otherwise indicated.

In embryos older than 20 hpf, *han^{s6}* mutants were identified based on their cardiac phenotype (*Yelon et al., 2000*). In younger embryos, PCR genotyping of *han^{s6}* mutants was conducted as previously described (*Yelon et al., 2000*), with the exception of embryos containing *Tg(hand2:EGFP)^{pd24}*, in which we used primers that amplify the first exon of *hand2* (5'– CCTTCGTACAGCCCTGAATAC – 3' and 5' – CCTTCGTACAGCCCTGAATAC – 3'). This exon is present in the wild-type allele but is absent in both *han^{s6}* homozygotes and the *TgBAC(hand2:EGFP)^{pd24}* transgene.

## Injection

Synthesis and injection of capped *hand2* mRNA was performed as described previously (*Schindler et al., 2014*). To knock down *hand2* function, we injected 3–6 ng of a previously characterized translation-blocking *hand2* morpholino at the one-cell stage (*Reichenbach et al., 2008*). To knock down *osr1* function, we used a previously characterized *osr1* translation-blocking start site morpholino, *osr1* ATG (*Tena et al., 2007*). 16–23 ng of this morpholino was injected with or without 1.6 ng of p53 morpholino (*Robu et al., 2007*).

## In situ hybridization

Standard and fluorescent whole-mount in situ hybridization were performed as previously described (*Brend and Holley, 2009*; *Thomas et al., 2008*), using the following probes: *atp1a1a.4* (ZDB-GENE-001212-4), *cdh17* (ZDB-GENE-030910-3), *efnb2a* (ZDB-GENE-990415-67), *etv2* (*etsrp*; ZDB-GENE-050622-14), *flt4* (ZDB-GENE-980526-326), *gata1* (ZDB-GENE-980536-268), *hand2* (ZDB-GENE-000511-1), *lhx1a* (*lim1*; ZDB-GENE-980526-347), *mrc1a* (ZDB-GENE-090915-4), *osr1* (ZDB-GENE-070321-1), *pax2a* (ZDB-GENE-990415-8), *slc12a3* (ZDB-GENE-030131-9505), and *tal1* (*scl*; ZDB-GENE-980526-501). To prepare an *osr1* probe, we amplified and subcloned the *osr1* cDNA using the primers 5' – GAGTTTCTACCCCGAGTAACCA – 3' and 5' – TTTTCAAAAATAAGTTTAAGGAA TCCA – 3'.

## Immunofluorescence and cell counting

Whole-mount immunofluorescence was performed as previously described (*Cooke et al., 2005*), using polyclonal antibodies against Pax2a at 1:100 dilution (Genetex, Irvine, CA, GTX128127) (RRID: AB_2630322), GFP at 1:1000 dilution (Life Technologies, Carlsbad, CA, A10262) (RRID: AB_2534023) or phospho-Histone H3 (Ser10) at 1:500 dilution (EMD Millipore Corporation, Temecula, CA, 05-1336-S) (RRID: AB_1977261), and the secondary antibodies goat anti-chick Alexa Fluor 488 (Life Technologies, A11039) (RRID: AB_2534096), goat anti-rabbit Alexa Fluor 488 (Life Technologies, A11008) (RRID: AB_143165), goat anti-mouse Alexa Fluor 546 (Life Technologies, A11003) (RRID: AB_2534071), goat anti-rabbit Alexa Fluor 594 (Life Technologies, A11012) (RRID: AB_10562717), or goat anti-rabbit Alexa Fluor 647 (Life Technologies, A21245) (RRID: AB_2535813), all at 1:100 dilution. Samples were then placed in SlowFade Gold anti-fade reagent (Life Technologies).

To count $Pax2a^+$ or $GFP^+$ cells in *Figures 2* and *7*, we flat-mounted and imaged embryos after dissecting away the yolk and the anterior portion of the embryo. In wild-type and $han^{s6}$ embryos, we examined a representative 250 μm long region in roughly the middle of the IM on one side of the embryo. This strategy allowed us to select contiguous regions that were unaffected by dissection artifacts. In *hs:hand2* embryos (*Figure 2*), we examined a 500 μm long region in order to minimize any selection bias that could be introduced from the variability associated with the relative dearth of $Pax2a^+$ cells. To count $Pax2a^+$ or $pH3^+$ cells in *Figure 2—figure supplement 2*, we chose to examine 400 μm long regions in undissected embryos, due to the low frequency of $Pax2a^+$ $pH3^+$ cells. In all cases, positive cells were determined through examination of both three-dimensional reconstructions and individual optical sections. Cell counts in *Figure 2*, *Figure 2—figure supplement 2*, and *Figure 7* are presented as the average number of positive cells per 100 μm. Statistical analysis of data was performed using Microsoft Excel to conduct unpaired *t*-tests.

## Histology

Histological analysis was performed on tissues embedded in Spurr Low-Viscosity embedding mixture. Embryos were fixed in 2% paraformaldehyde and 2.5% glutaraldehyde in 0.1 M sodium phosphate buffer (pH 7.2), post-fixed with 2% osmium tetroxide, dehydrated in an ethanol gradient, infiltrated with a propylene oxide/resin gradient, and then embedded. Samples were oriented as desired and incubated for 24 hr at 60° C for polymerization. 2–4 μm sections were cut using glass blades, collected on standard slides, and stained with 1% toluidine blue.

Transverse sections from 24 hpf embryos were examined at three different anterior-posterior levels along the pronephron, with two sections analyzed per level in each embryo. We processed three embryos per genotype. Cell counting and area measurements were performed using Zeiss ZEN software. Data variance homogeneity and normal distribution were confirmed using IBM SPSS Statistics 22 software.

## Imaging

Bright-field images were captured with a Zeiss Axiocam on a Zeiss Axiozoom microscope and processed using Zeiss AxioVision. Images of histological sections were captured using a Zeiss AxioImager A2 microscope coupled to a Zeiss Axiocam camera. Confocal images were collected by a Leica SP5 confocal laser-scanning microscope and analyzed using Imaris software (Bitplane, Switzerland).

## Replicates

All assessments of phenotypes and expression patterns were replicated in at least two independent experiments with comparable results. Embryos were collected from independent crosses, and experimental processing (injection, heat shock, and/or staining) was carried out on independent occasions. Two exceptions to this include data presented in *Figure 1Q–T* and *Figure 8C–E*. In each of those cases, multiple embryos were processed, and the n is reported in the associated figure legends.

## Acknowledgements

We thank members of the Yelon lab, N Chi, L Oxburgh, and D Traver for valuable discussions; I Drummond, K Poss, S Sumanas, J Torres-Vázquez, and D Traver for providing reagents; L Rincón-Camacho with assistance with histology; and H Knight for assistance with graphics.

# Additional information

## Funding

| Funder | Grant reference number | Author |
|---|---|---|
| California Institute for Regenerative Medicine | TG2-01154 | Elliot A Perens |
| A.P. Giannini Foundation | Postdoctoral Fellowship | Elliot A Perens |
| Universidad de los Andes | FAPA | Zayra V Garavito-Aguilar |
| Vicerrectoria de Investigaciones | P14.160422.007/01 | Zayra V Garavito-Aguilar |
| Colciencias Convocatoria | 617-2013-Joven Investigador Fellowship | Karen T Peña |
| National Institutes of Health | R01HL069594 | Deborah Yelon |
| March of Dimes Foundation | 1-FY16-257 | Deborah Yelon |
| National Institutes of Health | R01HL108599 | Deborah Yelon |

The funders had no role in study design, data collection and interpretation, or the decision to submit the work for publication.

## Author contributions

EAP, Conception and design, Acquisition of data, Analysis and interpretation of data, Writing the manuscript; ZVG-A, Conception and design, Acquisition of data, Analysis and interpretation of data, Input on the manuscript; GPG-V, KTP, Acquisition of data, Input on the manuscript; YLS, Contributed unpublished reagents, Input on the manuscript; DY, Conception and design, Analysis and interpretation of data, Writing the manuscript

## Author ORCIDs

Elliot A Perens, http://orcid.org/0000-0003-3377-7708

Zayra V Garavito-Aguilar, http://orcid.org/0000-0001-5671-7017

Gina P Guio-Vega, http://orcid.org/0000-0003-2156-6209

Karen T Peña, http://orcid.org/0000-0002-6543-5254

Yocheved L Schindler, http://orcid.org/0000-0002-4388-9511

Deborah Yelon, http://orcid.org/0000-0003-3523-4053

## Ethics

Animal experimentation: All zebrafish work followed protocols (S09125) approved by the University of California, San Diego IACUC.

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
