## [Decision Letter]

Thank you for submitting your article "Hand2 inhibits kidney specification while promoting vein formation within the posterior mesoderm" for consideration by *eLife*. Your article has been favorably evaluated by Marianne Bronner as the Senior Editor and three reviewers, including Melissa H. Little (Reviewer #3) and a member of our Board of Reviewing Editors.

The reviewers have discussed the reviews with one another and the Reviewing Editor has drafted this decision to help you prepare a revised submission

Summary:

In this manuscript from the Yelon laboratory, the authors address an important, yet understudied aspect of kidney biology, namely how the intermediate mesoderm (IM) is specified in the medio-lateral axis. They show, using loss- and gain-of-function approaches, that the bHLH transcription factor Hand2 limits the size of the embryonic kidney by restricting IM dimensions. Further analyses indicate that *hand2*, expressed laterally adjacent to the IM, functions in opposition to the zinc-finger transcription factor *Osr1* to balance the formation of the kidney and vein progenitors.

The data are convincing and the writing very clear. This is the first description of a role for Hand2 in delineation of the IM domain and in regulation of venous development along the embryo.

Essential revisions:

1) Figure 2: analyze the number of Pax2a^+^ cells in *hs:hand2* embryos.

2) The paper reports a genetic interaction between *hand2* and *osr1*, both of which are expressed laterally to the IM. This could be further explored: Is *Osr1* expression affected in *hand2* mutants? Is Hand2 expression affected in *osr1* morphants? Does *hand2* affect IM gene expression through its effects on *Osr1*? If these genes do affect the expression of each other, what is the molecular mechanism?

3) If any data using the mouse *hand2* mutant are available, they would add to the significance/impact of the findings. Specifically, would it be possible to examine the mutant mouse to look for either a widening of the IM or even the mesonephric duct based on *lhx1* and *pax2* expression or a hypoplasia of the metanephroi mid-gestation?

4) The data indicates that Hand2 represses IM formation. How does this happen molecularly? Does *hand2* directly repress key IM genes such as *pax2a* or *lhx1a*, or is it through a more indirect mechanism?

Suggested revision:

While the inference is that the role of Hand2 identified here will extend to patterning of the IM in other species, this is not investigated at all. Hence, the speculation that regulation of Hand2 will be important in regenerating the kidney from stem cells is quite an ambitious statement to make. It would be nice if the analysis could extend to mouse. Indeed, a number of manuscripts look at the effect of Hand2 mutation or overexpression, focusing on the jaw, limbs and enteric neurons, but never noting a change in the kidney. It is possible that this reflects a mild renal hypoplastic phenotype in mice that has been missed or an effect specifically on the pronephros or mesonephros but not the metanephros. We encourage you to explore this possibility if you are able to do so in a reasonable period of time as it would measurably increase the influence of this work. However, if this is not feasible, we will not insist on your completing the additional analysis for this publication.

---

## [Author Response]

*[…] Essential revisions:*

*1) Figure 2: analyze the number of Pax2a^+^ cells in hs:hand2 embryos.*

As requested, we have added new data to our manuscript in order to indicate the status of Pax2a^+^ cells in *hand2*-overexpressing embryos. Specifically, we have included new documentation of *pax2a* expression and new quantitation of the number of Pax2a^+^ cells in *hs:hand2* embryos. Both of these sets of data indicate that overexpression of *hand2* leads to a profound loss of the Pax2a^+^ population. In order to accommodate the addition of these data, we have modified Figure 2 and added a new figure supplement (Figure 2—figure supplement 1). We have also adjusted the bar graph shown in Figure 7 so that the units on its y-axis are comparable to those in the new bar graph shown in Figure 2. Finally, in addition to modifying the corresponding figure legends, we have adjusted the text of our Results section (subsection “*hand2* limits intermediate mesoderm dimensions by repressing intermediate mesoderm formation") and our Materials and methods section (subsection “Immunofluorescence and cell counting”) to include these new analyses.

*2) The paper reports a genetic interaction between hand2 and osr1, both of which are expressed laterally to the IM. This could be further explored: Is Osr1 expression affected in hand2 mutants? Is Hand2 expression affected in osr1 morphants? Does hand2 affect IM gene expression through its effects on Osr1? If these genes do affect the expression of each other, what is the molecular mechanism?*

As requested, we have added new data to our manuscript in order to examine the effects of *hand2* and *osr1* on each other's expression in the lateral mesoderm. These results appear to indicate normal expression of *osr1* in *han^s6^* mutants, as well as normal expression of *hand2* in *osr1* morphants. Therefore, our data do not support a model in which these genes affect each other's expression. Instead, our findings are consistent with a model in which *hand2* and *osr1* function in parallel genetic pathways that act in opposition to each other. These new data are presented in a new figure supplement (Figure 8—figure supplement 1) and are reflected in new text within our Results section (last paragraph) and in the corresponding figure legend.

*3) If any data using the mouse hand2 mutant are available, they would add to the significance/impact of the findings. Specifically, would it be possible to examine the mutant mouse to look for either a widening of the IM or even the mesonephric duct based on lhx1 and pax2 expression or a hypoplasia of the metanephroi mid-gestation?*

Please see our response below to the "Suggested Revision".

*4) The data indicates that Hand2 represses IM formation. How does this happen molecularly? Does hand2 directly repress key IM genes such as pax2a or lhx1a, or is it through a more indirect mechanism?*

We agree that it will be valuable for future studies to delve into the molecular mechanism through which Hand2 represses IM formation, and it will be particularly interesting to assess whether Hand2 acts directly or indirectly to repress IM genes. However, we feel that resolution of this issue would require extensive experimentation, including the creation of new reagents, and is therefore beyond the scope of our current study. In order to test the hypothesis that Hand2 acts directly to repress *pax2a* or *lhx1a*, we would need to evaluate whether binding of Hand2 to these genes’ regulatory regions is required to repress their expression. Performing ChIP-seq studies would require generation of a tagged knock-in allele of *hand2*, since no robust, ChIP-appropriate antibodies are currently available for Hand2. (The Zeller laboratory has recently taken this tagged-allele approach to study Hand2 targets in mouse (Osterwalder et al., 2014, Dev. Cell 31:345).) After identifying Hand2-binding sites, we would need to mutate those sites via genome editing in order to test whether they mediate repression of target gene expression. This will be an excellent future project, but we feel that the timeframe for generating reagents and performing the appropriate experiments would extend beyond that of a standard revision. In our revised manuscript, we have expanded our Discussion section to cover this issue. Specifically, our revised text notes that it is plausible that Hand2 could act directly to repress IM genes, since the Zeller laboratory’s data include Hand2-binding peaks associated with *Pax2* and *Lhx1a* in mouse (Osterwalder et al., 2014). Our revised text also mentions that Hand2 has been shown to influence transcription through both DNA-binding-dependent and DNA-binding-independent mechanisms (e.g. Funato et al., 2009, Development 136:615; Liu et al., 2009, Development 136:933), so Hand2 could have a “direct” influence on repression of IM gene transcription without necessarily engaging a Hand2-binding site in the corresponding regulatory regions. Finally, our revised text indicates that future studies will be needed to distinguish between direct and indirect models for Hand2 function in this context.

*Suggested revision:*

*While the inference is that the role of Hand2 identified here will extend to patterning of the IM in other species, this is not investigated at all. Hence, the speculation that regulation of Hand2 will be important in regenerating the kidney from stem cells is quite an ambitious statement to make. It would be nice if the analysis could extend to mouse. Indeed, a number of manuscripts look at the effect of Hand2 mutation or overexpression, focusing on the jaw, limbs and enteric neurons, but never noting a change in the kidney. It is possible that this reflects a mild renal hypoplastic phenotype in mice that has been missed or an effect specifically on the pronephros or mesonephros but not the metanephros. We encourage you to explore this possibility if you are able to do so in a reasonable period of time as it would measurably increase the influence of this work. However, if this is not feasible, we will not insist on your completing the additional analysis for this publication.*

The reviewers suggest, both in Essential Revision #3 and in the Suggested Revision, that data depicting the IM and/or kidney phenotypes in the mouse *Hand2* mutant would enhance the impact of our results, although the phrasing of the Suggested Revision indicates that the reviewers “will not insist on” our inclusion of mouse data in our revised manuscript. We agree that analysis of the conservation of the role of Hand2 during kidney development will be an important future endeavor. Unfortunately, no data regarding the relevant mouse phenotypes are currently available in the literature, and we have not performed independent studies of mouse *Hand2* mutants. In future work, we plan to conduct a comprehensive analysis of the appropriate tissues in mouse *Hand2* mutants, studying the dimensions of the IM and the development of the kidneys and urinary tract over a range of stages to insure that we can uncover any relevant phenotypes, even if they are relatively subtle. We note that analysis of IM derivatives at later stages may require use of a conditional *Hand2* allele, since the *Hand2* knockout is embryonic lethal (Srivastava et al., 1997, Nat. Genet. 16:154). Additionally, it may be necessary to analyze *Hand2;Hand1* double mutants, since both mouse genes are expressed in posterior mesoderm (Thomas et al., 1998, Dev. Biol. 196:228), whereas zebrafish have only a single Hand gene. Since we have not yet initiated these mouse experiments (and do not yet have the appropriate mice in our animal facility), we view this as a separate body of work that lies beyond the scope of our current manuscript (and beyond the timeframe of a standard revision). We are therefore hopeful that the text of the Suggested Revision accurately reflects the reviewers’ perspective. In our revised manuscript, we have extended our Discussion section (sixth paragraph) to indicate the importance of future investigation of mouse mutants and to comment on how prior investigations have not yet noted kidney phenotypes.